# Integrated Child and Family Hub models for detecting and responding to family adversity: protocol for a mixed-methods evaluation in two sites

Teresa Hall,[1] Sharon Goldfeld,[1] Hayley Loftus,[1] Suzy Honisett,[1] Hueiming Liu ![ORCID] ,[2] Denise De Souza,[3] Cate Bailey,[4] Andrea Reupert,[5] Marie B H Yap,[6] Valsamma Eapen ![ORCID] ,[7,8] Ric Haslam,[9] Lena Sanci,[10] Jane Fisher ![ORCID] ,[11] John Eastwood,[12] Ferdinand C Mukumbang ![ORCID] ,[13] Sarah Loveday,[1] Renee Jones,[1] Leanne Constable,[1] Suzie Forell,[14] Zoe Morris,[5] Alicia Montgomery,[15] Glenn Pringle,[16] Kim Dalziel,[17] Harriet Hiscock ![ORCID] [1]

For numbered affiliations see end of article.

**Correspondence to**
Dr Harriet Hiscock;
harriet.hiscock@rch.org.au

## ABSTRACT

**Introduction** Integrated community healthcare Hubs may offer a 'one stop shop' for service users with complex health and social needs, and more efficiently use service resources. Various policy imperatives exist to implement Hub models of care, however, there is a dearth of research specifically evaluating Hubs targeted at families experiencing adversity. To contribute to building this evidence, we propose to co-design, test and evaluate integrated Hub models of care in two Australian community health services in low socioeconomic areas that serve families experiencing adversity: Wyndham Vale in Victoria and Marrickville in New South Wales.

**Methods and analysis** This multisite convergent mixed-methods study will run over three phases to (1) develop the initial Hub programme theory through formative research; (2) test and, then, (3) refine the Hub theory using empirical data. Phase 1 involves co-design of each Hub with caregivers, community members and practitioners. Phase 2 uses caregiver and Hub practitioner surveys at baseline, and 6 and 12 months after Hub implementation, and in-depth interviews at 12 months. Two stakeholder groups will be recruited: caregivers (n=100–200 per site) and Hub practitioners (n=20–30 per site). The intervention is a co-located Hub providing health, social, legal and community services with no comparator. The primary outcomes are caregiver-reported: (i) identification of, (ii) interventions received and/or (iii) referrals received for adversity from Hub practitioners. The study also assesses child, caregiver, practitioner and system outcomes including mental health, parenting, quality of life, care experience and service linkages. Primary and secondary outcomes will be assessed by examining change in proportions/means from baseline to 6 months, from 6 to 12 months and from baseline to 12 months. Service linkages will be analysed using social network analysis. Costs of Hub implementation and a health economics analysis of unmet need will be conducted. Thematic analysis will be employed to analyse qualitative data.

**Ethics and dissemination** Royal Children's Hospital and Sydney Local Health District ethics committees

## STRENGTHS AND LIMITATIONS OF THIS STUDY

⇒ The 'Child and Family Hub' study is the first to our knowledge to co-design, test and evaluate two integrated Hub models of care for improving the detection of and response to family adversity for children aged 0–8 years.

⇒ Implementation and evaluation in two community health services across two Australian states will maximise the generalisability of findings for different community and service contexts.

⇒ The interdisciplinary conceptual framework underpinning this study strengthens its empirical robustness through synthesis of approaches from many disciplines to capture the complexity of the Hub models.

⇒ The study has no comparison group which means that any observed changes for primary and secondary outcomes cannot be directly attributed to the Hub models of care.

⇒ The English language inclusion criterion for participants limits our ability to engage culturally and linguistically diverse caregiver and practitioners.

have approved the study (HREC/62866/RCHM-2020). Participants and stakeholders will receive results through meetings, presentations and publications.
**Trial registration number** ISRCTN55495932.

## INTRODUCTION

### Family adversity and its negative impact on mental health

Interventions aimed at reducing or preventing the impact of adversity on children could avert a substantial proportion of the population burden of mental illness, and build on evidence demonstrating the health and economic benefits of investing early in life.[1] Family adversity includes a

range of adverse childhood experiences (ACEs) such as childhood maltreatment (eg, physical, verbal or sexual abuse), household dysfunction (eg, parental mental illness, family substance abuse), community dysfunction (eg, witnessing physical violence, discrimination), peer dysfunction (eg, stealing, bullying) and socio-economic deprivation.[2] While family adversity can negatively impact a range of physical and mental health outcomes, its negative effects on children's mental health are well established, increasing the risk of anxiety, depression and suicidality in childhood and across the life course.[3–8] Further, exposure to multiple ACEs predicts greater odds of poorer mental versus physical health outcomes in adult life.[9] Experiences of family adversity also negatively affect child's neurological and physiological development and educational outcomes,[10] which in turn lead to poor life-long health and mental health outcomes.[11]

### Detection and response to family adversity

As part of an Australian Centre of Research Excellence in Childhood Adversity and Mental Health, we will develop, test and evaluate an approach to improving the detection of and response to adversity in children aged 0–8 years and their families. Effective detection of adversity is crucial to the provision of appropriate, timely and evidence-based responses to the effects of adversity. However, major service barriers exist to the detection of, and responses to, adversity, including service time pressures, a lack of practitioner time, training and confidence in how to effectively discuss adversities, and practitioners' concerns about causing harm.[5 12–14] Family-level barriers also inhibit service access, including cognitive (ie, knowledge, awareness of services), psychological (ie, fear, shame and distrust of services), financial and structural barriers (ie, service availability, lengthy wait times, transport).[15–17] These service access barriers are amplified for families experiencing multiple adversities who require supports from health, social, education and community service systems. These systems are often siloed which is likely to contribute to the low utilisation of services by these families.[18 19] Hence, effective detection of, and responses to, adversity require intervention across multiple strata of intersectoral service systems.

### Integrated care for families experiencing adversity

Integrated care has the potential to overcome service fragmentation and respond to a range of holistic health and social needs for families and their children experiencing adversity. Integration is defined as care that: 'connect[s] the healthcare system (acute, primary medical and skilled) with other human service systems (eg, long-term care, education and vocational and housing services) to improve outcomes (clinical, satisfaction and efficiency)'.[20] Integrated care initiatives can increase uptake and ongoing engagement with child health services,[21 22] improve child mental health outcomes[23 24] and offer a cost-effective and acceptable service response for families.[25 26] Integrated community healthcare Hubs have

recently gained traction globally, driven by their appeal to offer a 'one stop shop' for service users with complex health and social needs, and to efficiently use service resources.[27] While there is no single definition of a Hub, the term is often used to describe a centralised service that offers a range of co-located, integrated services from multiple sectors, with linkages to external services for community-based supports.[27 28] Emerging evidence exists for outcomes of Hub models of care. For example, the Healthy Homes and Neighbourhoods (HHAN) initiative in Sydney, New South Wales (NSW), is a multi-component cross-agency care coordination network with a centralised Hub nested within a broader place-based initiative and has been shown to increase service access for families experiencing adversity.[29 30]

Various policy imperatives have been established to implement and evaluate Hub models of care for child and family mental health in Australia. These include the 2021 Royal Commission into Victoria's Mental Health System (recommending implementation and evaluation of 'infant, child and family health and well-being multidisciplinary community-based hubs'[31] and the 2021 National Children's Mental Health and Well-being Strategy.[32] The increased rates of child mental health problems resulting from the COVID-19 pandemic underscore the need for accessible, effective mental health support in communities.[33] However, there is a dearth of research specifically evaluating Hub models of integrated care for families experiencing adversity with a particular focus on child mental health. Extant research also does not specify when and how integration of care occurs within Hubs across the layers of a service ecological system (ie, clinical, professional, organisational and systems integration[31]), and which types of integration in co-located Hub models drive observed changes for children and families.

### The current study

To contribute to building this evidence, we propose to co-design, test and evaluate integrated Hub models of care in two Australian community healthcare services in low socioeconomic, metropolitan areas that serve families experiencing adversity, that is, Wyndham Vale in Victoria and Marrickville in NSW. Twenty-three per cent of children in Wyndham and 14.6% of children in Marrickville are developmentally vulnerable in one or more domains of the Australian Early Development Census.[34] In this study, we define a Hub as a co-located service providing an intersectoral health, social, legal and community response to family adversity. The Wyndham Vale site is a Hub in a local government area and the Marrickville site is a Hub nested within a broader place-based initiative. Co-design of a Hub at each site will involve caregivers, community members and service providers. The evidence generated from this study is crucial for increasing the conceptual precision around integrated Hub models of care and in addressing the evidence gap on their effectiveness for detecting and responding to family adversity and promoting positive child and caregiver mental health.

## Study aims and objectives

### Aim

1. To co-design, test and evaluate integrated Child and Family Hub models for detecting and responding to family adversity in children aged 0–8 years and their families in Wyndham Vale, Victoria, and Marrickville, NSW.
2. To formulate a realist-informed programme theory of how, why, for whom and under what conditions the Hubs work to detect and respond to family adversity.

### Objectives

1. To assess the impact of the Hub models on:

#### Primary outcomes

1.1. caregiver-reported (i) identification of, (ii) interventions received and/or (iii) referrals received for adversity from Hub practitioners.

#### Secondary outcomes

1.2. Hub practitioner-reported (i) identification of, (ii) interventions delivered and/or (iii) referrals provided for adversity,
1.3. Caregiver-reported uptake of referrals,
1.4. Child and caregiver mental health,
1.5. Infant temperament,
1.6. Child and caregiver global health,
1.7. Caregiver parenting,
1.8. Caregiver quality of life,
1.9. Caregiver personal well-being outcomes,
1.10. Caregiver experience of care,
1.11. Practitioner experience including confidence and competence in detecting and responding to adversity,
1.12. Linkages between intersectoral Hub practitioners, and
2. To determine the costs of implementing the Hub models, and
3. To determine the extent that the Hub models are providing the intended service option (ie, acceptability, feasibility and fidelity to the initial Hub programme theory).

Reporting in this protocol is based on the Strengthening the Reporting of Observational Studies in Epidemiology (STROBE) Statement.[35]

## METHODS AND ANALYSIS

This study is a multisite implementation and evaluation project that employs a convergent mixed-methods design conducted over three research phases displayed

**Phase 1**: Develop initial Hub theory (Oct 2020 – Aug 2021)

- Historical and current state needs assessment through online group discussions, interviews and meetings with intersectoral practitioners and caregivers
- Review and selection of focus CIMO configurations
- Co-design workshops and consultations to develop the Hub model

**Phase 2a**: Test initial Hub theory (Aug 2021 – Dec 2023)

Design: Convergent mixed-methods

Implementation, Context and Mechanism – data sources

- Caregiver baseline survey
- Hub practitioner baseline survey
- Process evaluation through routine data sources, observation, and Plan Do Study Act (PDSA) cycles
- In-depth interviews with Hub practitioners and caregivers

Outcomes

- Caregiver outcomes survey
- Hub practitioner outcomes survey
- In-depth interviews with Hub practitioners and caregivers
- Cost outcomes

Analytical approach

- Integrate data on context, mechanism and outcomes across multiple data sources
- Interpret outcomes data in reference to context and mechanism process data

**Phase 3**: Refine Hub model through local knowledge translation (Dec 2022-Dec 2023)

- Validate Hub theory through ongoing knowledge translation activities (e.g., forums, workshops, webinars)
- Refine Hub model based on study findings

**Figure 1** Evaluation processes and phases. CIMO, context-intervention-mechanism-outcome.

in figure 1 below. Quantitative and qualitative data will be collected in parallel and integrated during data analysis and interpretation.[36]

## Interdisciplinary conceptual framework

The study is underpinned by an interdisciplinary conceptual framework drawn from implementation science, critical realism, improvement science and collaborative co-design paradigms.[37–44] Interdisciplinary research strengthens empirical robustness through synthesis of approaches from many disciplines, and has been shown to increase the uptake of research into policy and practice.[45] The approach and rationale for the interdisciplinary framework in order of importance for this study are:

▶ *Implementation science*: Given the focus of our inquiry on complex Hub interventions, key implementation science frameworks form the basis of our process and outcomes evaluation, including our framing and synthesis of context using the Consolidated Framework for Implementation Research (CFIR).[42]

▶ *Critical realism and realist evaluation*: We draw on elements of critical realism and realist evaluation, a form of theory-driven evaluation, to theorise and examine 'what works, for whom, under what circumstances and how'.[46–49] Realist evaluation assumes that interactions between context and intervention mechanisms produce the outcomes of an intervention.[47 48 50 51] We focus our inquiry on the key context-intervention-mechanism-outcome (CIMO) configurations discussed below. We complement realist evaluation methods with critical realism[52 53] to understand the historical aspects of each locality and how this history influences the design and workings of each new Hub. This overcomes a key shortcoming of realist evaluation, that is, under theorising the multiple levels of social reality surrounding each Hub.[54 55]

▶ *Improvement science*: We use improvement science methods to implement agile learning and testing cycles that will iterate the Hub models and embed learning systems into practice.[43]

▶ *Collaborative co-design:* We use these principles and methods to co-design the Hub models and promote stakeholder engagement throughout the research process.

Figure 1 displays the three study phases based on Pawson and Tilley's[46] realist evaluation framework: (1) Development of the initial Hub programme theory through formative research; (2) Testing of the initial Hub theory using empirical data; and (3) Refinement of the initial Hub theory.

## Study phase 1: development of initial Hub programme theory through formative research

The initial Hub programme theory was developed through an intensive formative research phase during which we collected contextual information and co-designed the

Hub models with intersectoral (eg, health, social, legal and education) practitioners and caregivers from each local community. This phase occurred from October 2020 to June 2021 in Victoria and January to August 2021 in NSW. First, a historical and current needs assessment of the health, social, legal and community contexts of each site was conducted using qualitative data from online group discussions, individual interviews and meetings with key stakeholders including intersectoral practitioners (ie, from health, social, legal and community sectors) and caregivers. We used CFIR[42] to capture and synthesise these data into barriers and enablers across the outer setting, inner setting, individual level, process and intervention components.

Second, we reviewed the HHAN programme theories and other relevant realist literature focused on integrated care for children and families experiencing adversity[23 30 56–59] to identify key CIMO configurations relevant to the Hubs. HHAN provided a useful starting point because our study builds on the HHAN model of care. For pragmatic reasons, we focus the process and outcome evaluation in this study on three key mechanisms and associated CIMO configurations in the initial programme theory. Namely, (1) trust between caregivers and the service/practitioners, and between practitioners, (2) knowledge acquisition and motivation of practitioners and (3) perceived benefits of collaboration by practitioners. Figure 2 displays the initial realist-informed programme theory. We then formulated the logic model displayed in figure 3 proposing how the Hub models will be investigated and are theorised to contribute to outcomes. The logic model includes how immediate outcomes, that is, increased detection and response to adversity are hypothesised to lead to our intended outcomes, including increased uptake of services and improved child and caregiver mental health and caregiver quality of life.

Third, the Hub models were then developed and refined through a collaborative co-design process with intersectoral practitioners, caregivers and children at each site. The co-design process focused on the client journey through each proposed Hub and the workforce capacities and infrastructure needed to support implementation of the Hub. In Wyndham Vale, the Hub model was co-designed through an intensive 10-week series of co-design workshops and consultations with local caregivers and practitioners using human-centred design processes.[37–39] In Marrickville, the Hub model will be co-designed through focus group discussions and workshops employing the Nominal Group Technique consensus method[60] to prioritise and develop implementation strategies. Different approaches to co-design are adopted that reflect the capacity, capabilities and preferences for engagement of the research team and co-design partners at each site.

The theorised Hub models comprise:

▶ *Family and community partnerships*: intentional creation and strengthening of connections between the Hub and caregivers, community groups and individuals.

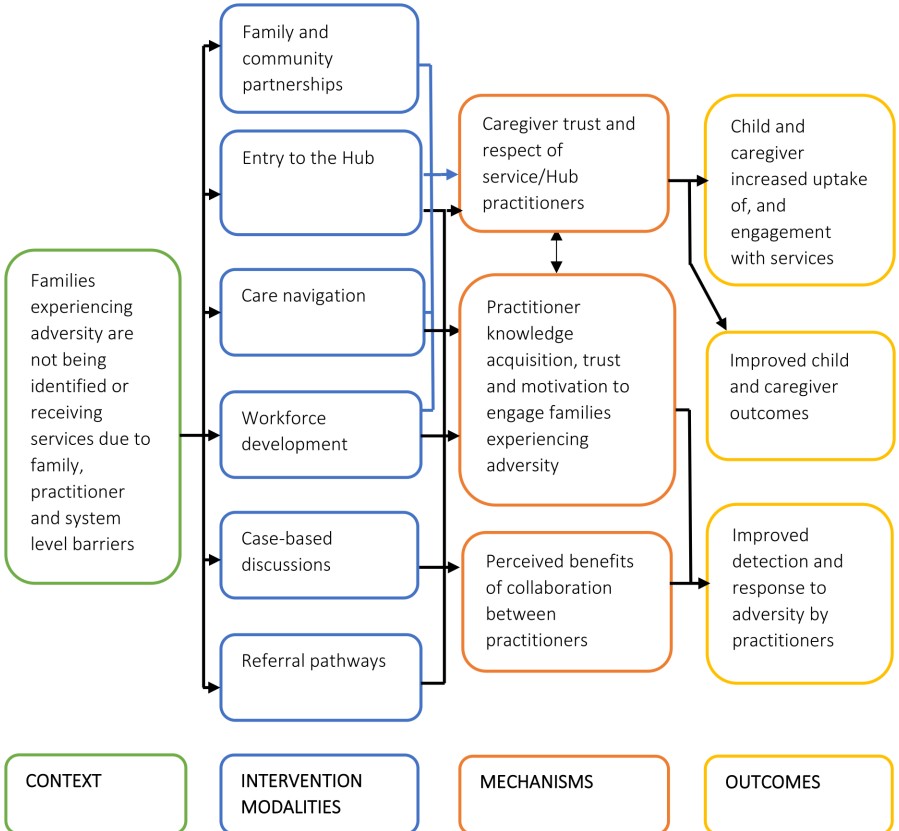

**Figure 2** Initial Hub programme theory from a realist perspective.

These partnerships also involve Hub practitioners working in partnership with families during clinical practice.

▶ *Entry to the Hub:* a 'no wrong door' approach in which caregivers are safely engaged in a conversation about adversity and provided with any necessary support and/or referrals regardless of how they enter the Hub.

▶ *Workforce development*: workforce capacity building and training of Hub practitioners to engage families in a

safe and respectful conversation to identify adversities and connect families to relevant support.

▶ *Case-based discussions:* monthly professional development with intersectoral Hub practitioners to embed training learnings into practice and facilitate between-practitioner referrals (ie, 'warm referrals').

▶ *Referral pathways into and out of the Hub*: systematic mapping of available health, community and social sector supports and services in the local area, linked

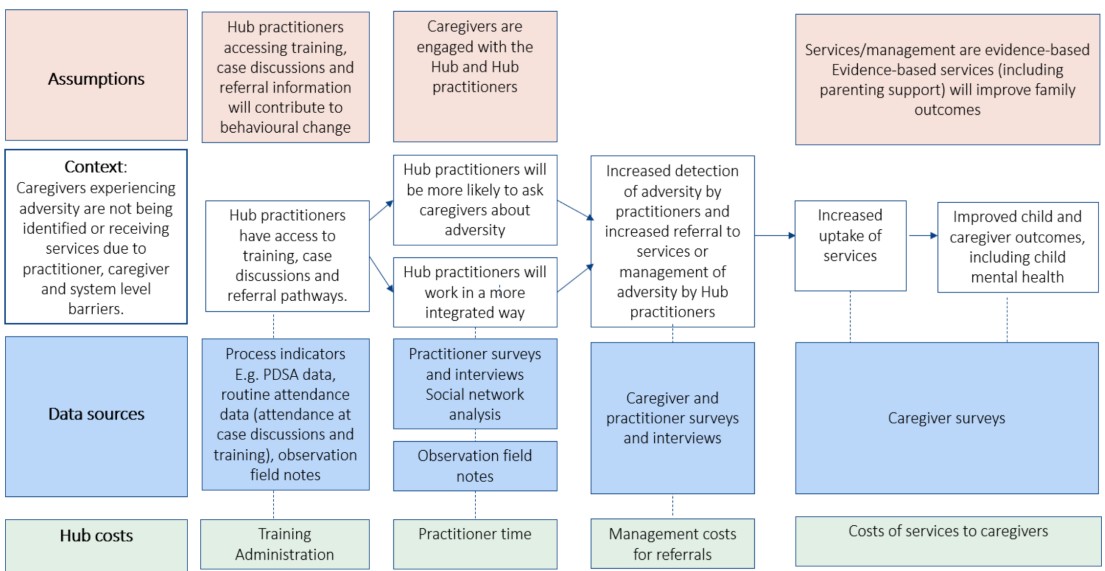

**Figure 3** Logic model proposing how the Hub models will be investigated. PDSA, Plan-Do-Study-Act.

to training of Hub practitioners to use this information in their practice.

▶ *Care navigation*: In Wyndham Vale, a Well-being Coordinator will support caregivers to identify the holistic needs of their child and/or family and assist them to navigate relevant services and supports in the community, social and health sectors. In Marrickville, care navigation is likely to include a Well-being Coordinator or a virtual application for caregivers that provides a risk stratification and recommendations of relevant services.

### Study phase 2: testing of initial Hub theory using empirical data

We will empirically test the initial programme theory for the Hubs through a mixed-methods process with 6 and 12 months outcomes evaluation. The evaluation aims to assess for whom, how and why the Hub models had an impact (if any) in the two different contexts of Wyndham Vale and Marrickville across the child, caregiver, practitioner and system level outcomes specified in figures 2 and 3, table 1 and 'Outcomes Assessment' below.

### Outcome evaluation
#### Design

The study is a mixed-methods repeated measures evaluation that uses caregiver and Hub practitioner surveys at baseline, 6 and 12 months after Hub implementation begins and in-depth interviews at 12 months.

#### Participants

We aim to recruit two stakeholder groups: caregivers (n=100–200 per site) and Hub practitioners (n=20–30 per site).

#### *Caregivers*

Potential caregiver participants are those who: (i) care for a child aged 0–8 years, including women who are pregnant, (ii) access any of the universal or specialist services provided in the Hub, such as general practitioners (GPs), paediatricians, lawyers and (iii) can understand written or spoken English language. The English language criterion is a limitation of our study; however, funding is not available to provide interpreters. Furthermore, culturally and linguistically diverse persons participated in an informed and meaningful way in phone-based or in-person surveys for HHAN (personal communication Professor Eastwood).

Given the preventative focus of this study, caregivers will not be screened for adverse experiences as an inclusion criterion. Instead, caregivers will report the frequency of 15 adversity types in the baseline survey. These adversity types are derived from the Parent Engagement Resource[61] and include challenges with: social support, finances, housing, employment, family physical health, parent mental health, parenting, child neglect, alcohol and substance use, family relationships, family violence, child abuse, visa and immigration issues, crime issues and discrimination.

#### *Hub practitioners*

Potential practitioner participants are those who: (i) work in any of the intersectoral services provided as part of each Hub, including GPs, maternal and child health nurses, paediatricians, allied health professionals, social workers, lawyers, and (ii) can understand written or spoken English language.

#### Sample size calculation

As this study is testing the Hub models using a non-experimental design, we have not conducted a formal sample size calculation. However, the sample size of 200–400 caregivers is likely to provide rich data on the primary and secondary outcomes.

#### Recruitment and consent

The study is designed to accommodate a range of participants, many of whom may have complex life circumstances.[62] Figure 4 presents the caregiver participant study flow. We will recruit caregivers from the waiting rooms of each Hub and through Hub practitioners who will ask their clients' permission for the research team to contact them. Researchers will approach potential caregiver participants (in the waiting room or on the telephone), show and/or email them an informational video and study information pack (including a written Caregiver Participant Information Consent Form, PICF) and invite them to take part in the study. Participants can provide informed consent verbally on the phone or in-person, or in writing by clicking through the online consent form preceding the survey. Caregivers will be provided with a AUD$25 honorarium for each survey and/or interview they complete.

We will invite practitioners working in each Hub who have been identified and recruited by the study team or health centre managers. We will send an invitation email to potential practitioner participants with the Practitioner PICF attached before they attend the workforce training. Practitioner participants will provide written (online or hard copy) informed consent prior to completing the baseline survey.

### Outcome assessment

The study will assess the child, caregiver, practitioner and systems outcomes summarised in table 1.

#### Data collection
##### *Caregiver and Hub practitioner surveys*

The Caregiver and Hub practitioner baseline and outcome surveys consisting of the measures outlined in table 1 and available in online supplemental file 1 will be built in Research Electronic Data Capture (REDCap)[63] hosted by Murdoch Children's Research Institute. After completing an informed consent process, caregivers can complete the baseline survey: (i) in paper form, (ii) over the telephone with a researcher or (iii) online. Practitioners will complete the practitioner baseline survey in paper form or online via REDCap. The practitioner survey will be sent towards the end of baseline data

**Table 1** Primary and secondary outcomes

| Primary outcomes | | Baseline assessment | | 6 and 12 months post Hub implementation begins | |
|---|---|---|---|---|---|
| | | Caregiver survey | Practitioner survey | Caregiver survey | Practitioner survey |
| *Caregiver* | | | | | |
| Identification of adversity | Increase in the proportion of caregivers who report being asked by a service provider about adversity in the past 6 months. | X | | X | |
| Intervention for adversity | Increase in the proportion of caregivers who report spending extra time with or receiving an intervention from a Hub service provider for adversity in the past 6 months. | X | | X | |
| Referrals for adversity | Increase in the proportion of caregivers who report receiving a referral to an intersectoral service for adversity in the past 6 months. | X | | X | |
| **Secondary outcomes** | | | | | |
| *Child* | | | | | |
| Infant temperament | Increase in the proportion of caregivers who report their infant is easier/much easier than average; assessed through single caregiver-reported item on infant temperament; has a moderate correlation ($r$=0.51) with the Easy-Difficult Scale (EDS) of Australian version short form of Revised Infant Temperament Questionnaire[78]; completed for one child* aged 0–8 in each family 0 to <2 years. | X | | X | |
| Child mental health | Decrease in the mean scores for caregiver-reported internalising or externalising symptoms for their child. Completed for one child* in each family. For children aged 0 to <2 years: Ages & Stages Questionnaire Social-Emotional Second Edition (ASQ-SE2).[79]† For children aged ≥2 to 8 years: Strengths & Difficulties Questionnaire.[80] | X | | X | |
| Global health | Increase in the mean scores of caregiver-reported general child health; assessed through single item (GHQ-S1) from Child Health Questionnaire[81]; completed for one child* aged 0–8 in each family. | X | | X | |
| *Caregiver* | | | | | |
| Uptake of referrals | Increase in the proportion of caregivers who report uptake of referrals to other services in the past 6 months. | X | | X | |
| Mental health | Decrease in the mean scores of caregiver-reported psychological distress as assessed by Kessler Psychological Distress Scale 6 (K6); 6-item.[82] | X | | X | |
| Global health | Increase in means scores of caregiver-reported general health; assessed through single item of the Short Form Health Survey (SF-12).[83] | X | | X | |
| Parental warmth, parenting hostility and efficacy | Increase in mean scores of parental warmth and efficacy; decrease in mean scores on parenting hostility. Three self-report subscales drawn from the Longitudinal Study of Australian Children (LSAC), Australia's first nationally representative longitudinal study of child development which will allow for national comparisons of the study data[84]; assessing parental warmth, parenting hostility and efficacy. Parental warmth (6-items), parenting hostility (5-items) and parenting efficacy (4-items). | X | | X | |
| Quality of life | Increase in mean caregiver quality of life scores calculated from the EuroQol Health and Well-being Instrument Short Form (EQ-HWB-S).[72] | X | | X | |
| Caregiver experience | Caregiver reported acceptability and feasibility of the Hub; Increase in the proportion of caregivers who report their satisfaction with Hub care as measured by three items from the Australia Bureau of Statistics Patient Experiences in Australia Survey.[85] | X | | X | |
| Personal well-being | Increase in mean scores of caregiver-reported personal well-being outcomes measured by the Personal Well-being Index; 7 items.[86] | X | | X | |
| *Practitioner* | | | | | |
| Identification of adversity | Increase in the proportion of practitioners who report asking about adversity in the past 6 months. | | X | | X |

Continued

**Table 1** Continued

| Primary outcomes | | Baseline assessment | 6 and 12 months post Hub implementation begins |
|---|---|---|---|
| Intervention for adversity | Increase in the proportion of practitioners who report spending extra time or providing an intervention for adversity in the past 6 months. | X | X |
| Referrals for adversity | Increase in the proportion of practitioners who report referring caregivers to an intersectoral service for adversity in the past 6 months. | X | X |
| Practitioner experience | Practitioner reported acceptability and feasibility of the Hub; Increase in the proportion of practitioners who report feeling confident and competent to detect and support families experiencing adversity. | X | X |
| *System level* | | | |
| Strength and structure of intersectoral service linkages | Increase in the number and strength of service linkages between Hub practitioners as assessed through social network analysis (SNA) indicators for network density, degree, centrality and betweenness, and map of linkages between Hub practitioners based on i) contact, (ii) referrals to, (iii) referrals from, and (iv) quality of the relationship.[68–71] | X | X |
| Health economics outcomes | Costs of implementation of the Hub models, caregiver-reported intersectoral service usage and value of unmet need. | X | X |

*Caregivers with more than one child will respond to questions pertaining to one child in their family based on the child they are most concerned about.
†The ASQ-SE is limited by its design as a screening tool that may not be a sensitive outcome measure. The measure is used in this study because it more directly measures mental health and well-being than the ASQ.

collection to minimise the potential impact of the survey on clinical practice prior to the Hub testing period. Caregiver and practitioner participants will be contacted by telephone, email or text message by the research team 6 and 12 months after the Hub implementation begins and invited to complete the follow-up survey.

*Caregiver and Hub practitioner interviews*
Twelve months after Hub implementation begins, we will conduct realist-informed semi-structured interviews with 18–24 caregivers and 5–10 Hub practitioners per site.[64] We will purposively sample participants to capture caregivers from diverse cultural backgrounds, experiences of adversities and varying levels of engagement with intersectoral services. These realist-informed interviews will enhance the testing of the initial programme theory to triangulate with the findings from the quantitative measures. The interviews aim to uncover (i) the mechanisms and contextual elements relevant to the Hub models (to inform theory development), (ii) acceptability and feasibility of the Hub, (iii) their experience of being asked about adversity, the care offered to them and the process of referrals including barriers and enablers to uptake (caregivers only) and (iv) confidence and competence in detecting and supporting families experiencing adversity (practitioners only).

Quantitative data analysis
Statistical analyses will be conducted using Stata or R statistical software packages.[65 66] The baseline characteristics of the caregivers, children and practitioners will be described using the mean, median and IQR for continuous data and proportions for categorical data. Primary and secondary outcomes will be assessed by examining change in proportions/means from baseline to 6 months, from 6 to 12 months and from baseline to 12 months. We will also conduct linear regression analyses to assess the association between each caregiver-reported primary outcome at 6 and 12 months with the outcomes of child and caregiver mental health and caregiver quality of life, at 6 and 12 months, respectively.

Quantitative measures of the strength and structure of service linkages between Hub practitioners will be analysed using social network analysis (SNA) software package UCINET.[67] SNA is a complex systems discipline and quantitative methodology widely used to measure networked relationships between organisations and individual actors in health service and policy settings.[68–71] UCINET[67] will calculate indicators (ie, density, degree, centrality and betweenness) for the network and each Hub practitioner based on: (i) contact, (ii) referrals to and from and (iii) quality of the relationship. These indicators and analytical approach are recommended for descriptive social network analysis.[68–71]

Health economics analysis
Costs of the Hubs will be identified, including practitioner training, administration for referrals, personnel to oversee clinical implementation and other relevant costs. Downstream costs of services and potential cost-offsets will be collected and analysed to obtain the cost

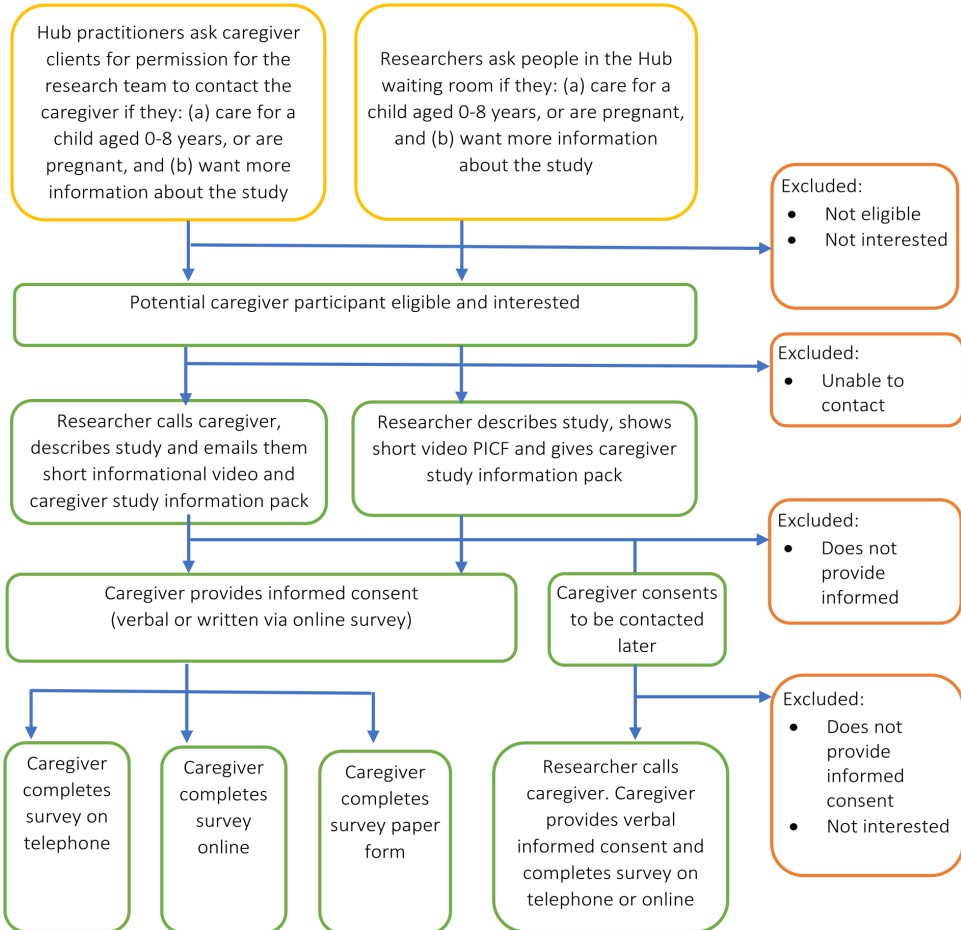

**Figure 4** Caregiver participant study flow. PICF, Participant Information Consent Form.

per additional child/or caregiver attending the Hub. Quality of life will be measured using the carer-specific measure (EQ-HWB-S[72]). An analysis of the value of unmet need will be conducted by determining whether (i) the child and family's needs were met, (ii) no needs or (iii) unmet needs using the Ages & Stages Questionnaire or Strengths and Difficulties Questionnaire (SDQ) to determine perceived need compared with the level of service use.

### Qualitative data analysis

Individual interviews will be transcribed verbatim by a transcription service and imported into NVivo Release 1.0[73] to assist in the process of analysis. Experienced qualitative researchers will employ Braun and Clarke's six-stage reflexive thematic analysis[74] to analyse the qualitative interview data. In line with the critical realist philosophy underpinning this study, we will employ inductive coding and deductive (a priori) coding based on the CIMO configurations for the Hub.

### Process evaluation and quality improvement cycles

We will conduct a process evaluation using multiple data sources, including routinely collected data (eg, attendance rates at training and visits to the Well-being Coordinator) and observation field notes of Hub practitioner clinical practice, the training sessions and the case-based discussions. The process evaluation examines the acceptability, feasibility and fidelity of implementation for each Hub component shown in the initial Hub programme theory in figure 2. The process evaluation will also include realist analytical methods in which observed outcomes are explained by looking into the mechanisms and contextual elements contributing to the focus CIMOs for this study.

We will also conduct improvement cycles guided by the Model for Improvement.[43] Small improvement teams will be formed at each site who will establish their specific improvement aims and measurements. Once the aims are agreed, the team will undertake short learning cycles via Plan-Do-Study-Act (PDSA) cycles.[43] Each PDSA cycle begins by articulating: the change and recording predictions about what we expect to happen (Plan); attempting to make the change and documenting what happened (Do); comparing the results to the predictions (Study); and then deciding on what to do next (Act). By the end of the project, learning cycles will be embedded as part of routine Hub service delivery, making the Hub more likely to be sustained and used.[41] To maximise rigour, we will triangulate data collected through the process evaluation and PDSA cycles with outcome measures collected through the 6 and 12 month surveys.[75]

## Study phase 3: refinement of initial Hub programme theory

We will use the process and outcome evaluation data to refine the initial Hub programme theory. Further testing will be undertaken by explicitly seeking to confirm or contradict theories throughout the PDSA cycles, in-depth interviews and knowledge translation activities (eg, forums, workshops and webinars). A key strength of our study is the assessment of outcomes across multiple levels of the Hub intervention (ie, child, family, practitioner, service and system levels). The triangulation of multiple data sources to evaluate the implementation of the Hub model serves to maximise the study's rigour and confidence in our findings.[75 76]

## Patient and public involvement

Caregivers and practitioners were actively involved in the co-design of the Hub models and are members of local advisory groups at each site which oversee the study design, recruitment, piloting of instruments, interpretation and dissemination of findings. Their engagement is aimed at ensuring the research is responsive to each context and to facilitate the translation of findings into practice.[77]

## ETHICS AND DISSEMINATION

The study protocol has been approved by the Royal Children's Hospital Human Research Ethics Committee (HREC/62866/RCHM-2020) and Sydney Local Health District (HREC/62866/RCHM-2020). All participants will be asked to provide written or verbal informed consent prior to completing the surveys and interviews. Each participant will be allocated a participant identification number that is kept securely under password protection accessible only to the research team.

## Confidentiality

Participant confidentiality is strictly held in trust by the participating investigators, research staff and the sponsoring institution and their agents. The study protocol, documentation, data and all other information generated will be held in strict confidence. No information concerning the study or the data will be released to any unauthorised third party other than Murdoch Children's Research Institute, without prior approval by the participant.

## Access to data

All investigators will be able to access cleaned study data for analysis. Data will be housed on Murdoch Children's Research Institute's secure network or via file transfer, and all data sets will be password protected. To ensure confidentiality, dispersed data files will not include identifying participant information.

## Managing participant distress

While we do not anticipate any major risks to participants, the surveys and individual interviews may raise issues that are uncomfortable, upsetting or frustrating for some participants. To reduce the potential for distress to participants, the study team will provide clear explanations about why the study is being conducted, how the information will be used and the kinds of questions that will be asked. If the participant becomes distressed at any point, the researchers will empathise with the participant, stop the survey or interview if requested and discuss ways in which they might get support from family, friends or formal support services.

## Dissemination

We will publish study findings in international peer-reviewed journals and present papers at national and international conferences. At a local level in Wyndham Vale and Marrickville, we will communicate project learnings to local stakeholders via presentations, community social media and research summaries. We will provide research summaries to local and state-wide media and in social media posts. A knowledge translation strategy will disseminate findings using a range of mediums including state-wide and national Hub networks, communities of practice and regular engagements with relevant government departments. The strategy is intended to support the scale up of effective components to other community health services in Victoria, NSW and across Australia.

**Author affiliations**
[1]Centre for Community Child Health, Murdoch Childrens Research Institute, Parkville, Victoria, Australia
[2]Centre of Health Systems Science, The George Institute for Global Health, Camperdown, New South Wales, Australia
[3]Centre for Research in Education and Sustainability, Torrens University Australia - Fitzroy Campus, Melbourne, Victoria, Australia
[4]Health Economics Unit, The University of Melbourne School of Population and Global Health, Melbourne, Victoria, Australia
[5]Department of Education, Monash University, Melbourne, Victoria, Australia
[6]Monash University, Clayton, Victoria, Australia
[7]ICAMHS, South Western Sydney Local Health District, Liverpool, New South Wales, Australia
[8]School of Psychiatry, University of New South Wales, Sydney, New South Wales, Australia
[9]Mental Health, The Royal Children's Hospital Melbourne, Parkville, Victoria, Australia
[10]Department of General Practice, University of Melbourne, Melbourne, Victoria, Australia
[11]Jean Hailes Research Unit, School of Public Health and Preventative Medicine, Monash University, Melbourne, Victoria, Australia
[12]Paediatrics and Child Health, School of Public Health, The University of Sydney, Sydney, New South Wales, Australia
[13]School of Public Health, University of the Western Cape, Cape Town, Western Cape, South Africa
[14]Health Justice Australia, Melbourne, Victoria, Australia
[15]Sydney Institute for Women, Children, and their Families, Sydney Local Health District, Sydney, New South Wales, Australia
[16]IPC Health, Wyndham Vale, Victoria, Australia
[17]School of Population and Global Health, The University of Melbourne, Carlton, Victoria, Australia

**Acknowledgements** We thank all families, practitioners and researchers who are working together on the Child and Family Hubs project, and members of the advisory and implementation groups at each site for their oversight and guidance. We acknowledge Professor Anthony Jorm as the chief investigator responsible for

the first stream of work (Theme A) conducted by the CRE Childhood Adversity and Mental Health.

**Contributors** HH, SG and JE are chief investigators responsible for conceptual development, study design and manuscript preparation. TH was involved in concept development, study design and drafted the manuscript. HLo, HLi, DDS, CB, AR, MBHY, VE, RH, LS, JF, JE, FCM, SL, RJ, LC, SF, ZM, AM, GP and KD were involved in concept development, study design and manuscript preparation, and approved the final manuscript.

**Funding** This research is supported by the Australian National Health and Medical Research Council and Beyond Blue grant number 1153419. Murdoch Children's Research Institute is supported by the Victorian Government's Operational Infrastructure Support Program. HH is supported by an NHMRC Practitioner Fellowship 1136222. SG is supported by an NHMRC Practitioner Fellowship 1155290. JF holds the Finkel Chair in Global Health which is supported by the Finkel Family Foundation.

**Competing interests** None declared.

**Patient and public involvement** Patients and/or the public were involved in the design, or conduct, or reporting, or dissemination plans of this research. Refer to the Methods section for further details.

**Patient consent for publication** Not applicable.

**Provenance and peer review** Not commissioned; externally peer reviewed.

**ORCID iDs**
Hueiming Liu http://orcid.org/0000-0001-9077-8673
Valsamma Eapen http://orcid.org/0000-0001-6296-8306
Jane Fisher http://orcid.org/0000-0002-1959-6807
Ferdinand C Mukumbang http://orcid.org/0000-0003-1441-2172
Harriet Hiscock http://orcid.org/0000-0003-3017-2770

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
