## [Reviewer comments · BMJ Open]

ARTICLE DETAILS

TITLE (PROVISIONAL)	Integrated Child and Family Hub models for detecting and responding to family adversity: protocol for a mixed-methods evaluation in two sites.
AUTHORS	Hall, Tess; Goldfeld, Sharon; Loftus, Hayley; Honisett, Suzy; Liu, Hueiming; De Souza, Denise; Bailey, Cate; Reupert, Andrea; Yap, Marie; Eapen, Valsamma; Haslam, Ric; Sanci, Lena; Fisher, Jane; Eastwood, John; Mukumbang, Ferdinand; Loveday, Sarah; Jones, Renee; Constable, Leanne; Forell, Suzie; Morris, Zoe; Montgomery, Alicia; Pringle, Glenn; Dalziel, Kim; Hiscock, Harriet

VERSION 1 – REVIEW

REVIEWER	Otago University, General Practice & Rural Health
REVIEW RETURNED	01-Aug-2021

GENERAL COMMENTS	This paper presents the study protocol for a mixed-methods evaluation of Integrated Child and Family Hub models for detecting and responding to family adversity. The study addresses an important health service evaluation area. The study protocol is well written, appropriately structured (using relevant study design reporting standards / checklists) and covers in appropriate detail all aspects of the study design. I do not consider it needs any revision.
--

REVIEWER	Mandy Allison University of Colorado Denver - Anschutz Medical Campus
REVIEW RETURNED	09-Jan-2022

GENERAL COMMENTS	Thank you for the opportunity to review this study protocol. I look forward to learning more about the results of the study as it progresses. I have included some suggestions to improve the clarity of the manuscript and some questions for the authors below. ABSTRACT: I understand that describing a complex study in a small amount of space is challenging. After reading the entire manuscript, I have some suggestions for how the abstract might be edited for clarity and to be more compelling, especially if a reader only looks at the abstract and does not read the entire protocol. 1) The co-design aspect of the protocol is a real strength. Consider including more information about WHO the co-designers were/are
---

and how they will be integrated in the study. Some of this could be done by providing a bit more information about the 3 phases of the study in the abstract as I suggest below.

2) I suggest either naming the conceptual frameworks used in the abstract or not referring to the conceptual frameworks in the abstract.

3) I found the description of the 3 research phases later in the manuscript to be very helpful for conceptualizing the study. I suggest very briefly describing the 3 phases in the abstract, even if it means editing to leave other aspects of the study design out (for example, 'congruent mixed methods study').

4) I suggest naming or providing categories for the 'child, caregiver, and practitioner' outcomes (page 6, line 9).

INTRODUCTION

5) Consider moving the last sentence of the first paragraph (page 7, starting at line 24) to the beginning/first sentence of the introduction. This sentence clarifies the sentences that precede it.

6) The section about the 'current study' mentions co-design but doesn't explicitly state who the co-designers are. Since the co-design is a strength of the study, I suggest briefly describing the co-design aspects in more detail here.

7) Page 9, line 19: I'm not sure what is meant by 'flow-on' here? Consider re-wording.

METHODS

8) Figure 1 was helpful for conceptualizing the study. Consider referring to the figure and providing an overview at the very beginning of the Methods section.

9) Page 13: Why were two different co-design methods used in the two different settings? Was that the choice of the community? I don't necessarily think it is a problem that two different methods were used, but I would like to know WHY two different methods were used.

10) The use of in-depth, qualitative interviews at 12 months should be noted in the abstract--it helps clarify the mixed methods design.

11) Page 14, 'Sample size calculation': it makes sense that this study does not involve a formal sample size calculation; however, can the authors provide additional information about each of the communities where the study is taking place? What are the demographic and social characteristics (that are available through census data for example) of these communities? How many people/families are expected to interact with the Hub in some way in each community?

12) The table on page 20 notes that families will answer questions about child mental health for one child in the family. How will that child be chosen?

13) The ASQ-SE is going to be used as an outcome measure. I understand that other studies have used the ASQ-SE as an outcome measure due to the lack of validated measures of socio-emotional health in children under age 18 months to 2 years old; however, the ASQ-SE is designed as a screening tool and isn't necessarily a sensitive outcome measure. This should be explained and/or included as a limitation.

14) Will any sort of monetary stipend or honorarium be provided to study participants? If yes, please describe how much and to whom. If no, please note that.

VERSION 1 – AUTHOR RESPONSE

- 1) We have specified that phase 1 of the research “involves the co-design of the Hub at each site with family, community and service providers”. (page 2, lines 10-13)
- 2) We have removed reference to the conceptual framework in the abstract for clarity in line with Reviewer 2’s suggestion. (page 2, line 12)
- 3) We have elaborated on the three research phases in the abstract as described later in the manuscript. We have kept ‘convergent mixed methods study’ as this is the best practice reporting requirement for mixed methods research. (page 2, lines 10-13)
- 4) We have included the high-level categories for the child, caregiver, practitioner and systems outcomes in the abstract. (page 2, line 19-20)

- 5) Thank you for this suggestion, we have made the change which helps with paragraph flow and clarity. (page 3, line 10-12)
- 6) We have specified that the co-design process will involve “caregivers, community members and service providers’ (page 4 line 31-32). We specify more details about the co-design later in the manuscript under the Phase 1 so do not agree that more details of the co-design at this point in the manuscript are necessary (page 7 lines 21-30).
- 7) We agree that ‘flow on’ is ambiguous. We have reworded this sentence for simplicity. (page 4 line 35)
- 8) We have revised to refer to Figure 1 at the start of the Methods section. (page 5 line 25)
- 9) We have revised to state the reasons for the different approaches to co-design at each site as follows: “Different approaches to co-design are adopted that reflect the capacity, capabilities and preferences for engagement of the research team and co-design partners at each site.” (page 7 line 29-30)
- 10) The abstract already states that in-depth interviews will be conducted at 12 months. As such, we have made no changes. (page 2 line 14)
- 11) We have provided additional information about Wyndham and Marrickville communities including that they are both “low socioeconomic, metropolitan areas” and that “Twenty-three percent of children in Wyndham and 14.6% of children in Marrickville are developmentally vulnerable in one or more domains of the Australian Early Development Census.” (page 4 lines 27-30). Given the reduced numbers of families attending services as a result of Covid, we cannot reliably predict how many families will attend the Hubs. We hope numbers will grow as the Hubs becomes established in each site.
- 12) We have added a note to the table to clarify the following: “Caregivers with more than one child will respond to questions pertaining to one child in their family based on the child they are most concerned about.” (page 14, lines 1-2)
- 13) Thank you for highlighting this use of the ASQ-SE. We have added this as a limitation in the notes section of the table when the ASQ-SE is first introduced (page 14, line 4).
- 14) Caregivers will be provided with a \$25 honorarium for each survey and/or interview they complete. These details have been added to the Recruitment and consent section (page 10 line 24-25).